# Uncontrolled Two-Step Iterative Calibration Algorithm for Lidar–IMU System

**DOI:** 10.3390/s23063119

**Published:** 2023-03-14

**Authors:** Shilun Yin, Donghai Xie, Yibo Fu, Zhibo Wang, Ruofei Zhong

**Affiliations:** College of Resource Environment and Tourism, Capital Normal University, Beijing 100048, China; shilunyin@cnu.edu.cn (S.Y.);

**Keywords:** SLAM, lidar–IMU calibration, lidar, IMU

## Abstract

Calibration of sensors is critical for the precise functioning of lidar–IMU systems. However, the accuracy of the system can be compromised if motion distortion is not considered. This study proposes a novel uncontrolled two-step iterative calibration algorithm that eliminates motion distortion and improves the accuracy of lidar–IMU systems. Initially, the algorithm corrects the distortion of rotational motion by matching the original inter-frame point cloud. Then, the point cloud is further matched with IMU after the prediction of attitude. The algorithm performs iterative motion distortion correction and rotation matrix calculation to obtain high-precision calibration results. In comparison with existing algorithms, the proposed algorithm boasts high accuracy, robustness, and efficiency. This high-precision calibration result can benefit a wide range of acquisition platforms, including handheld, unmanned ground vehicle (UGV), and backpack lidar–IMU systems.

## 1. Introduction

Accurate state estimation of a platform is a critical requirement for autonomous driving. While satellite navigation systems can provide accurate positioning information in the wilderness, their signal loss is commonplace in metropolitan areas, limiting their usefulness for autonomous driving applications. The Simultaneous Localization and Mapping (SLAM) method, which enables autonomous localization and mapping, has emerged as a research hotspot in the fields of autonomous driving and robot navigation [1,2,3,4]. Depending on the type of data used, SLAM can be classified into two main categories: visual SLAM and lidar SLAM.

Visual SLAM was the dominant focus of early research. Traditional methods were primarily implemented through filtering [5], and classical algorithms were utilized to accomplish localization and mapping through the Extended Kalman Filter (EKF) [4]. Subsequently, keyframe-based visual SLAM techniques matured, and mapping algorithms such as ORB-SLAM became popular [6]. While visual sensors can more accurately extract environmental features when operating at low speeds in texture-rich areas, their accuracy decreases with increasing motion speed. Thus, multi-sensor fusion SLAM techniques, particularly those that combine cameras and IMUs (VI-SLAM), have emerged. VI-SLAM is a low-cost combination of camera and IMU, which has gained a lot of attention. Experiments have demonstrated that VI-SLAM can operate effectively in complex scenes with violent motions [7]. Cameras can capture rich scene details, while IMUs can be used to predict the platform’s attitude and position. The two sensors complement each other to yield better positioning accuracy and mapping results. The VI-ORB algorithm has been proposed based on the ORB-SLAM algorithm [8]. The VINS-Mono algorithm employs a tightly coupled, non-linear optimization-based approach for scale recovery through a monocular camera and IMU, demonstrating better robustness and generality [9]. The PL-VIO algorithm proposes a tightly coupled monocular visual inertial odometry system based on the use of point and line features [10], and the VINS-Fusion algorithm extends VINS-Mono to integrate stereo cameras and GPS devices, enhancing the applicability of VINS-Mono [11].

In recent years, lidar SLAM has grown rapidly, thanks to the reduction in hardware costs of lidar. The robustness of lidar to light and viewpoint changes and its high range accuracy make it an important sensor for localization and mapping in large indoor and outdoor environments. The principle of lidar SLAM is similar to visual SLAM, and the LOAM algorithm [12] has been a representative algorithm for lidar SLAM. Compared to previous methods, the LOAM algorithm is innovative in reducing the number of point clouds by feature point extraction, improving matching accuracy, and constructing optimized cost functions based on point-to-line and point-to-face distances for different feature point types. Several extended algorithms based on LOAM have been proposed, including A-LOAM [13], LeGO-LOAM [14], LIO-SAM [15], and F-LOAM [16]. These algorithms offer unique features such as a simplified version of LOAM, separation of ground points, frame–local map matching, and optimization strategies. These extended algorithms have shown improved operating efficiency and are suitable for localization and mapping in different scenes.

Multi-sensor fusion has emerged as a promising approach to improving the performance of lidar SLAM, particularly in challenging environments. By integrating lidar data with complementary sensor information, such as IMU data, the accuracy of localization and mapping can be significantly improved. A key challenge in lidar and IMU data fusion is the calibration of external parameters between the two sensors. Sensor calibration is crucial in the field of autonomous driving, especially in determining the vehicle’s attitude, which refers to its orientation with respect to the ground, such as pitch, roll, and yaw angles. The vehicle’s attitude is essential in determining its position, trajectory, and behavior on the road, such as lane-keeping, turning, and braking. Therefore, it is essential to calibrate the sensors to reduce their errors and improve their accuracy and consistency [17,18].

Temporal calibration is a crucial component of multi-sensor calibration. Since the lidar and IMU have different emission frequencies, the system may produce time deviations, necessitating pre-synchronization correction. GNSS hardware calibration is a popular method for optimizing this process [19]. However, software-based calibration methods are also available [20,21,22].

The widely used calibration methods are offline calibration or online calibration through markers [23]. However, these methods can be complicated and time-consuming. Recently, Zhejiang University (ZJU) proposed an online calibration method LI-Calib that does not require markers, which employs continuous-time trajectory equations and optimal estimation to decompose space into individual units and identify planar elements for data association [24]. Despite its potential, this method may have higher data requirements in practice. LI-Init is a robust, real-time initialization method for lidar–inertial systems proposed by Hong Kong University (HKU) [25]. The proposed method calibrates the temporal offset and extrinsic parameter between lidars and IMUs. However, this method is more suitable for complex lidar–IMU systems. Since this method involves SLAM mapping, it consumes a lot of resources. For simpler lidar–IMU systems it requires a long processing time.

We present an uncontrolled two-step iterative calibration algorithm for the lidar–IMU system to address the challenge of calibrating the rotation matrix between the lidar and IMU. Our algorithm leverages the high accuracy of the IMU during short periods and corrects the rotational distortion of a single frame of a point cloud acquired in one scan cycle, while also predicting the rotation matrix between adjacent point cloud frames based on the IMU’s angle values. At the start of the calibration process, the rotation matrix is unknown, hence, we employ a two-step approach. In the first step, we match the point cloud without aberration correction to obtain an initial calibration matrix. In the second step, we correct the distortion of the point cloud caused by rotational motion and match the point cloud again to obtain a higher accuracy calibration matrix. The calibration matrix is updated iteratively until convergence. Our proposed algorithm is effective in calibrating the rotation matrix for lidar–IMU systems, and can significantly improve the accuracy of localization and mapping.

## 2. Materials and Methods

### 2.1. Related Work

#### 2.1.1. Calibration Principle for Lidar–IMU System

As illustrated in Figure 1, two distinct methods exist to transform a point $p_{k+1}^{imu}$ in the IMU coordinate system at time k+1 to the lidar coordinate system at time k. The first involves transforming the point to the lidar coordinate system at time k+1 and then to the lidar coordinate system position at time k based on the relative position attitude of the lidar. The second involves first transforming the point to the IMU coordinate system position at time k and then to the lidar coordinate system position at time k based on the relative position attitude of the IMU. 

In Equation (1), the transformation of the point $p_{k+1}^{imu}$ is shown in two ways as mentioned earlier.
(1)$$R_{k,k+1}^{lidar}R_{lidar,imu}p_{k+1}^{imu}{=R}_{lidar,imu}R_{k,k+1}^{imu}p_{k+1}^{imu}$$

This equation involves the rotation matrices $R_{k,k+1}^{lidar}$, $R_{lidar,imu}$, and $R_{k,k+1}^{imu}$. Simplified versions of these matrices are shown in Equations (2) and (3), where $q_{l}^{b}$ is the quaternion of rotation from the lidar to the IMU coordinate system, $q_{b_{k+1}}^{b_{k}}$ is the result of IMU pre-integration measured from moment $b_{k}$ to moment $b_{k+1}$, and $q_{l_{k+1}}^{l_{k}}$ represents the attitude change at moment $l_{k+1}$ with respect to the lidar coordinate system at moment $l_{k}$.
(2)$$R_{k,k+1}^{lidar}R_{lidar,imu}{=R}_{lidar,imu}R_{k,k+1}^{imu}$$
(3)$$q_{b_{k+1}}^{b_{k}}{=q}_{l}^{b}⊗q_{l_{k+1}}^{l_{k}}⊗q_{b}^{l}$$

The quaternion multiplication can be represented as the multiplication of matrices with quaternions [19], as shown in Equations (4)–(7), where $Q_{1}^{+}$ is the left multiplication matrix and $Q_{2}^{-}$ is the right multiplication matrix. $Q_{b_{k+1}}^{b_{k}+}$ is the left multiplication matrix of the result of the attitude pre-integration of the IMU measurements during moment $b_{k}$ to moment $b_{k+1}$, and $Q_{l_{k+1}}^{l_{k}-}$ is the right multiplication matrix of the attitude change of moment $b_{k+1}$ with respect to the lidar coordinate system of moment $b_{k}$ matrix. After merging, these equations can be transformed into Equation (8). By assuming that there are n sets of measurement data for attitude calibration, the system of chi-square linear equations shown in Equation (9) can be established and solved by the singular value decomposition (SVD) method to obtain the value of $q_{l}^{b}$ [26].
(4)$$q1⊗{q2=Q}_{1}^{+}q2\to Q_{1}^{+}{=q}_{w}I+\left[\begin{array}{cc} 0 & {-q}_{v}^{T}\\ q_{v} & {\left[q_{v}\right]}_{\times } \end{array}\right]$$
(5)$$q1⊗{q2=Q}_{2}^{-}q1\to Q_{2}^{-}{=q}_{w}I+\left[\begin{array}{cc} 0 & {-q}_{v}^{T}\\ q_{v} & -{\left[q_{v}\right]}_{\times } \end{array}\right]$$
(6)$${\left[q_{v}\right]}_{\times }=\left[\begin{array}{ccc} 0 & {-q}_{z} & {-q}_{y}\\ q_{z} & 0 & {-q}_{x}\\ {-q}_{y} & q_{x} & 0 \end{array}\right]$$
(7)$$q=\left[\begin{array}{cc} q_{w} & q_{v} \end{array}\right]$$
(8)$$\left(Q_{b_{k+1}}^{b_{k}+}{-Q}_{l_{k+1}}^{l_{k}-}\right)q_{l}^{b}=0$$
(9)$${\left[\begin{array}{c} Q_{b_{1}}^{b_{0}+}{-Q}_{l_{1}}^{l_{0}-}\\ Q_{b_{2}}^{b_{1}+}{-Q}_{l_{2}}^{l_{1}-}\\ \ldots \ldots \\ Q_{b_{n}}^{b_{n-1}+}{-Q}_{l_{n}}^{l_{n-1}-} \end{array}\right]}_{4n\times 4}q_{l}^{b}{=A}_{4n\times 4}q_{l}^{b}=0$$

#### 2.1.2. Matching Algorithms for Point Clouds

In this study, various point cloud matching algorithms are investigated and compared for calculating the relative rotation and translation between two adjacent frames. The algorithms considered include the Iterative Closest Point (ICP) [27], Generalized ICP (GICP) [28], Normal Distributions Transform (NDT) [29], and OMP-NDT [30]. The ICP algorithm aims to minimize the distance between the two point clouds by finding the spatial transformation of the overlapping area, which is simple and does not require segmentation or feature extraction of point clouds. However, its computational cost for finding the nearest corresponding point is high, and it lacks the use of point cloud structure information. The GICP algorithm improves on the ICP algorithm by comprehensively considering point-to-point, point-to-surface, and surface-to-surface strategies, resulting in higher accuracy and robustness. The NDT algorithm divides the point cloud space into voxel grids and calculates the normal distribution of the point cloud in each grid, providing fast computation and less influence from moving targets in the scene. The OMP-NDT algorithm is an OpenMP-based parallel NDT algorithm derived from PCL that uses a multi-threaded approach for faster alignment speed, with a small deviation in matching results. The voxel nearest neighbor search in this study was performed using the DIRECT 7 algorithm.

### 2.2. Uncontrolled Two-Step Iterative Calibration Algorithm

Figure 2 illustrates the proposed two-step calibration algorithm. In Step 1, the initial rotation matrix is calculated based on the matching of the original point cloud and the integration of IMU data. The pre-integrated attitude measurement values of the IMU from time $b_{k}$ to $b_{k+1}$ are obtained using the angular velocity data. A system of homogeneous linear equations is established based on Equation (9), and the SVD method is applied to solve for the value of $q_{l}^{b}$. In Step 2, the initial calibration result is used to perform motion distortion correction on the original point cloud. Attitude prediction is made based on the IMU integral, and the corrected point cloud is matched and calibrated again to obtain a more accurate calibration result. The process is iterated until convergence is achieved.

In the motion distortion correction process, each point in a scan period’s point cloud data is transformed into a reference coordinate system $c_{0}$, which has its origin typically located at the start of the scan period. The rotation and translation of each point’s coordinate system $c_{k}$ with respect to $c_{0}$ is determined, allowing the transformation of $p_{k}$ from $c_{k}$ to $c_{0}$. Since the 6-axis IMU only outputs angular velocity and linear acceleration, the correction of motion distortion caused by attitude is considered. The three attitude angles corresponding to each coordinate system $c_{k}$ can be obtained from the integration of angular velocities measured by the IMU. Figure 3 illustrates this principle.

To correct motion distortion in the lidar–IMU system, we first evaluate the rotation matrices $R_{ax}$, $R_{ay}$, and $R_{az}$ around the x, y, and z axes, respectively, by using the angular velocities $\omega_{xi}$, $\omega_{yi}$, and $\omega_{zi}$ measured by the 6-axis IMU at each point i. Equation (10) is used to calculate the rotation matrix for each axis, and then Equation (11) is used to determine the transformation matrix $R_{0,k}$ from $c_{k}$ to $c_{0}$. The motion distortion is then solved using Equation (12). The correction process is related to the initial calibrated rotation matrix and can be further optimized through iterations. By utilizing the rotation matrix obtained from these equations, we can predict the transformation in the subsequent frame and improve the speed and accuracy of the matching process.
$$R_{a_{x}}=\overset{k}{\underset{i=1}{\sum }}\omega_{xi}$$
(10)$$R_{a_{y}}=\overset{k}{\underset{i=1}{\sum }}\omega_{yi}$$
$$R_{a_{z}}=\overset{k}{\underset{i=1}{\sum }}\omega_{zi}$$
(11)$$R_{0,k}{=R}_{a_{x}}R_{a_{y}}R_{a_{z}}$$
(12)$$p_{k}^{ud}{=R}_{0,k}p_{k}$$

## 3. Results

### 3.1. Experimental Data 

In order to evaluate the proposed method, point cloud data were collected from five distinct locations and datasets, including Rotation, Park, Campus, Walk, and Cnu, using high-precision equipment. The Rotation, Park, Campus, and Walk data are publicly available datasets collected using a Velodyne VLP-16 lidar and a Microstrain 3DM-GX5-25 IMU at the Massachusetts Institute of Technology (MIT) [15]. Meanwhile, the Cnu dataset was obtained at Capital Normal University using a Velodyne VLP-16 lidar and an Inertial Labs INS-D IMU. The specific details of each dataset are outlined in Table 1. To obtain the position and attitude information of each lidar frame, we utilized the Simultaneous Localization and Mapping (SLAM) algorithm. As shown in Figure 4, the resulting stitched point cloud represents the collected data from each location. To compare the performance of the proposed method, we calculated the error by converting the estimated rotation matrix to Euler angles and comparing them to the ground truth values. The ground truth calibration values of Rotation, Park, Campus, and Walk datasets are reported as (0, 0, 0), as established in previous research [15]. On the other hand, for the Cnu dataset, the ground truth calibration values were established offline in the laboratory using control points and measured at (−179.90, −0.35, −88.54).

### 3.2. Comparative Analysis of Calibration Accuracy of Different Processing Methods

In this study, the accuracy of the calibration is evaluated by decomposing the rotation matrix between the calibrated and actual values into three Euler angles (roll angle, pitch angle, heading angle), and calculating the distance error (in degrees) between the three angles. The smaller the distance error, the higher the calibration accuracy. In Equation (13), $\Delta_{r}$, $\Delta_{p}$, and $\Delta_{y}$ represent the differences in roll, pitch, and heading angles, respectively. To assess the effectiveness of different calibration methods, we compared four approaches, three of which were processed following the steps outlined in Figure 2 with only slight variations between them. The first method (Method I) utilized the original point cloud without motion distortion correction, IMU prediction, or iterative processing, thus eliminating the frame-to-local map matching. The second (Method II) and third (Method III) methods were novel techniques introduced in this study, with Method III differing from Method II only in the inclusion of IMU prediction. Finally, we compared our approach to LI-Calib proposed by ZJU [24] and LI-Init proposed by HKU [25]. By comparing the results of these four calibration methods, we aim to determine the most effective approach for lidar–IMU calibration.
(13)$$Rotation\_error=\sqrt[{2}]{\Delta_{r}{}^{2}+\Delta_{p}{}^{2}+\Delta_{y}{}^{2}}$$

The first three calibration methods utilize the OMP-NDT algorithm for point cloud frame matching, as illustrated in Figure 5. Method II proposed in this paper showed an overall improvement in accuracy after incorporating motion distortion correction and iterative processing. However, the introduction of IMU prediction did not consistently improve the accuracy and, in some cases, led to a decrease in accuracy. This variability in results may be attributed to the accuracy of the IMU measurements. The current study did not account for the drift phenomenon of IMU hardware over time during IMU integration, which may impact the accuracy of the predictions and overall matching results.

Although LI-Calib provides open-source software with a GUI, the format of the point cloud used by LI-Calib is different from our experimental data, and only the Park dataset can be processed at present. To compare the accuracy of LI-Calib with our proposed method, we conducted three tests in the same environment and evaluated the stability of the algorithms. The results presented in Table 2 show that our proposed Method II has an average error of 11.01°, which is stable and consistent across all three tests. On the other hand, the average error of LI-Calib is 36.63°, and it exhibits larger fluctuations. The intermediate results of LI-Calib open-source software are shown in Figure 6.

In order to evaluate the performance of the proposed Method II, we conducted experiments on Rotation data and compared it with the LI-Init method. Our results demonstrate that Method II outperforms the LI-Init method in terms of calibration accuracy, with slightly better results and more efficient running time. These findings are consistent with our expectations, and support the potential of Method II to address calibration issues in a more accurate and time-effective manner. A detailed presentation of the experimental results can be found in Table 3.

### 3.3. Calibration Accuracy of Different Point Cloud Matching Algorithms 

In this study, the impact of various point cloud matching algorithms on the calibration accuracy is investigated, as point cloud matching quality is a crucial factor in calibration. Four representative matching algorithms, including ICP, NDT, GICP, and OMP-NDT, are selected to evaluate their impact on calibration accuracy. As presented in Table 4, calibration results vary among the different matching algorithms. The effects of the matching algorithms on the calibration accuracy can be clearly seen in Figure 7, which demonstrates that the accuracy of calibration results varies for different data. Overall, the calibration accuracy is substantially improved by using GICP or NDT, while ICP exhibits inferior performance in comparison.

### 3.4. Analysis of the Influence of the Scenario on the Calibration Accuracy

In order to assess the performance of the calibration methods and matching algorithms, we analyzed the angular change between adjacent frames of the first 100 frames of point clouds for each scene. This was carried out by calculating the angular distance based on the matching results, and the results are visualized in Figure 8a. Furthermore, we evaluated the calibration errors of the five datasets, which were acquired using different calibration methods and matching algorithms. The results are shown in Table 5.

In this study, we evaluated the calibration errors of different scenes using a handheld lidar–IMU combination system for fixed position acquisition, as well as a combined system using a backpack. Our results indicate that the average calibration error for Rotation data was only 3.58°, which is a relatively small overall error. However, the errors for Park and Walk data were higher, while Campus data had smaller errors, suggesting that calibration errors are influenced by the scene type. The average error for the combined lidar–IMU system using the backpack (Cnu data) was 9.09°, which was higher than that of the handheld system. Figure 8b shows a scatter plot of the mean angular variation of the different data versus the mean calibration error. Our findings suggest that maintaining a steady point cloud matching throughout calibration and having a large rotational change between neighboring point cloud frames can result in higher calibration accuracy. To achieve this, we recommend collecting calibration data while the equipment is uniformly rotating in the same location to minimize the impact of equipment movement on matching. Additionally, including wall structures in the collected scene can improve the effect of extracting stable point cloud features, which can lead to a regular point cloud distribution and improve point cloud matching.

## 4. Discussion

The findings reveal that the calibration accuracy is significantly affected by the data acquisition method and point cloud matching algorithm employed. The calibration results of the GICP and NDT algorithms are superior to those of the conventional ICP algorithm, indicating that advanced matching algorithms can enhance the calibration performance of the lidar–IMU system. Furthermore, the calibration outcomes of different scenarios show that the calibration results of Rotation data are the most stable and have higher accuracy, while the errors of the Park and Walk scenarios are comparatively larger. These results emphasize the importance of carefully selecting the data acquisition method and point cloud matching algorithm for achieving reliable calibration results.

In addition, this study recommends using handheld lidar, keeping the data acquisition in a single fixed location, and keeping the rotation speed constant and reasonable to guarantee stable calibration results. Accurate calibration and successful matching can both be achieved by gathering regular wall information in the scenario.

## 5. Conclusions and Future Work

This study introduces an uncontrolled two-step iterative calibration algorithm to address the current calibration issues with the lidar–IMU system. The proposed algorithm directly utilizes the data acquired by the lidar–IMU system to obtain the initial rotation matrix through initial matching and then iteratively increases the matching accuracy by exploiting motion distortion and IMU pre-integration. The experimental comparison demonstrates that the method in this study can produce significantly better calibration results for vehicle-mounted, handheld, backpack, and other lidar acquisition modalities when compared to the conventional method. In comparison with existing algorithms, the proposed algorithm boasts high accuracy, robustness, and efficiency.

In our future work, we aim to further improve the calibration algorithm by increasing the number of experimental scenes, examining the impact of frame-to-submap matching on calibration accuracy, and accounting for IMU data inaccuracy. We also aim to enhance the open-source code and employ GPU to accelerate point cloud matching, thereby enabling the calibration program to operate more efficiently and conveniently. We also plan to investigate the challenge of achieving non-hardware time synchronization between the lidar and IMU sensors in our system. As each sensor has its own timestamp, this can be a complex task. We aim to leverage the results obtained through pose synchronization to develop an algorithm for time synchronization. By integrating the pose and time synchronization algorithms, we expect to improve the overall accuracy and robustness of our system. 

## Figures and Tables

**Figure 1 sensors-23-03119-f001:**
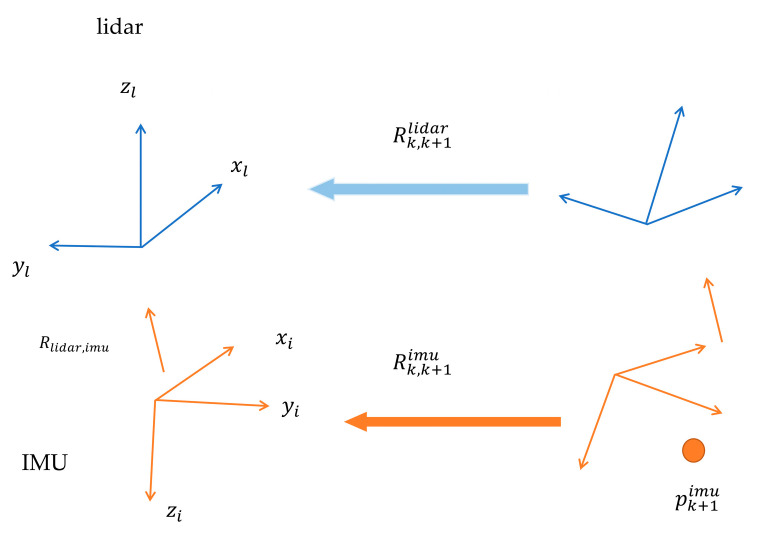
Calibration schematic for lidar–IMU system.

**Figure 2 sensors-23-03119-f002:**
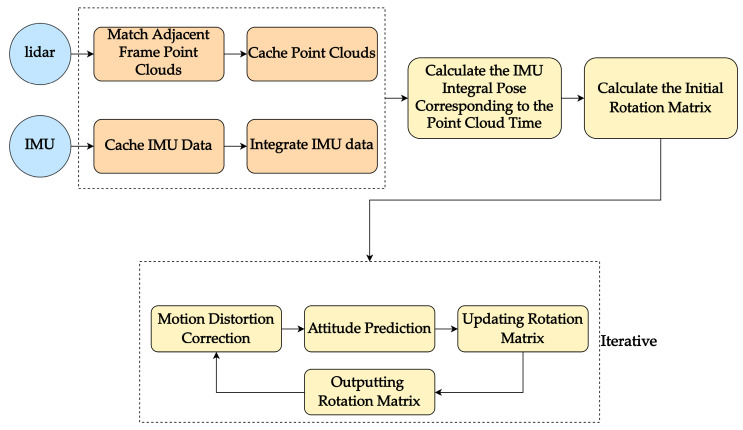
Flowchart of the two-step calibration algorithm.

**Figure 3 sensors-23-03119-f003:**
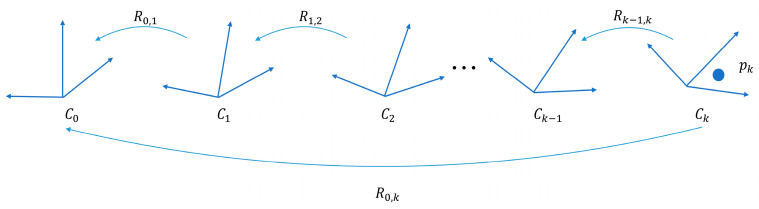
Principle of motion distortion correction.

**Figure 4 sensors-23-03119-f004:**
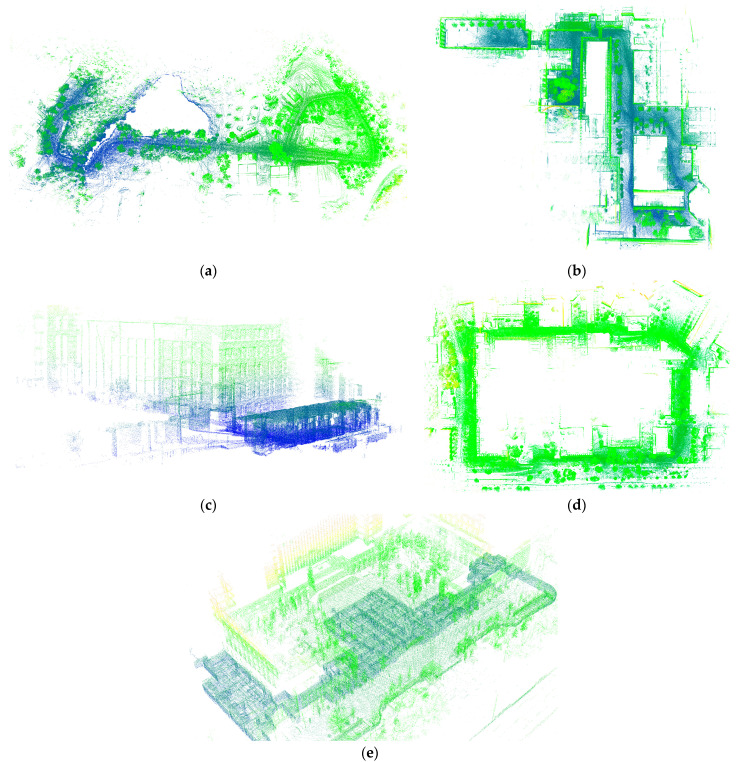
The results obtained by applying SLAM algorithm to different data: (**a**) Park data; (**b**) Walk data; (**c**) Rotation data; (**d**) Campus data; (**e**) Cnu data.

**Figure 5 sensors-23-03119-f005:**
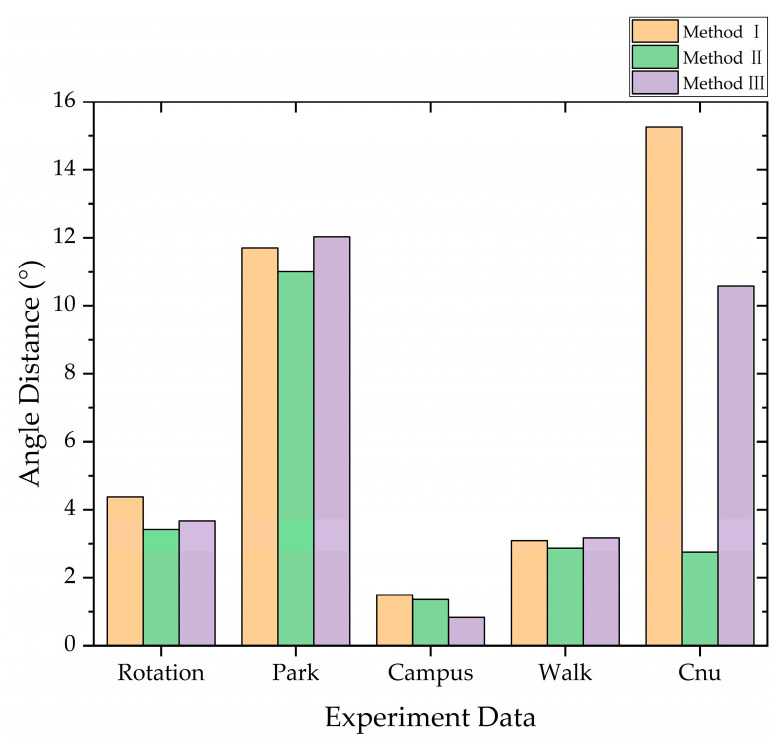
Calibration error of three methods.

**Figure 6 sensors-23-03119-f006:**
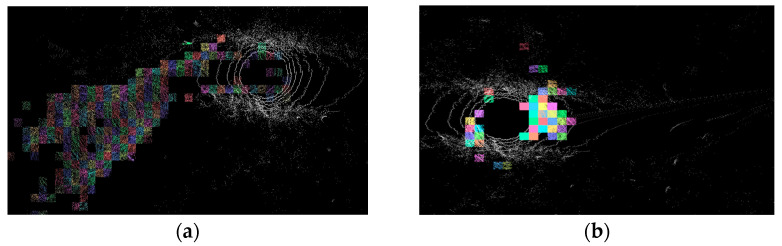
(**a**,**b**) Operation results using ZJU’s algorithm.

**Figure 7 sensors-23-03119-f007:**
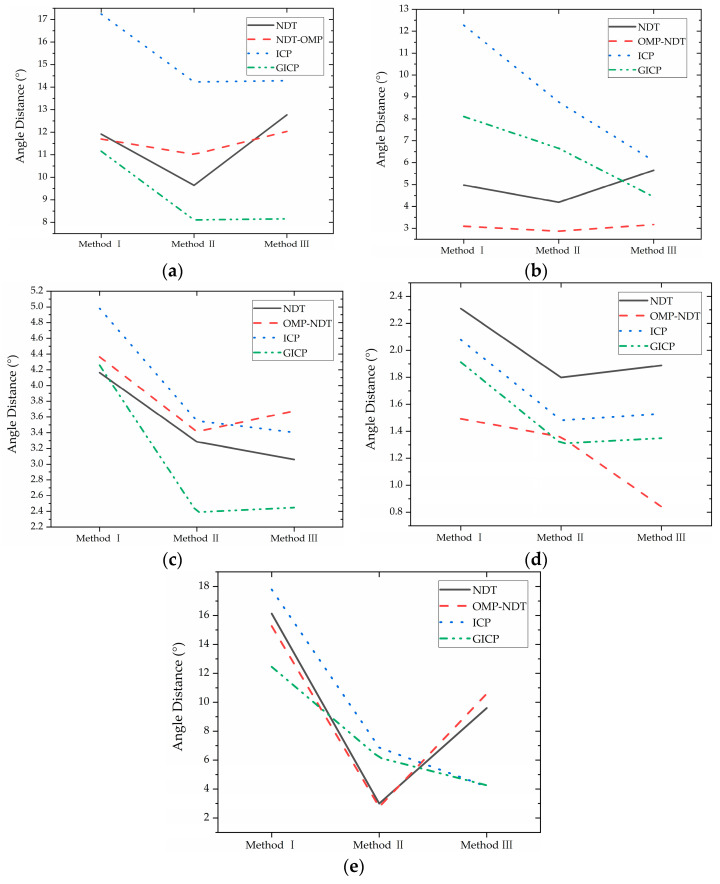
Calibration results of different matching algorithms for different data: (**a**) Park data; (**b**) Walk data; (**c**) Rotation data; (**d**) Campus data; (**e**) Cnu data.

**Figure 8 sensors-23-03119-f008:**
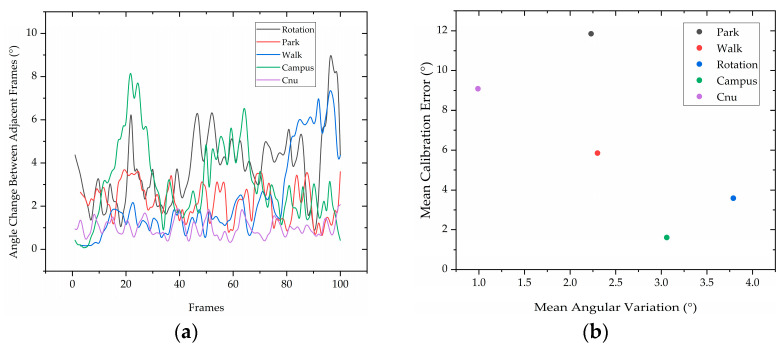
(**a**) Angle change between adjacent frames of different data; (**b**) the relationship between angular variation of different data and calibration error.

**Table 1 sensors-23-03119-t001:** Experimental data information.

Name	Duration (Seconds)	Acquisition Platform	Location
Rotation	58.60	Handheld	MIT
Park	560.00	Vehicle-mounted	MIT
Campus	994.00	Handheld	MIT
Walk	655.00	Handheld	MIT
Cnu	1199.20	Backpack	Capital Normal University

**Table 2 sensors-23-03119-t002:** Comparison of the accuracy of Method II and LI-Calib.

Name	Method II (°)	LI-Calib (°)
Park	11.01	18.74
	11.01	53.09
	11.01	38.05
Average error	11.01	36.63

**Table 3 sensors-23-03119-t003:** Comparison of the accuracy of Method II and LI-Init.

Method	Rotation (°)	Time (s)
Method II	3.41	40.2
LI-Init	4.08	47.3

**Table 4 sensors-23-03119-t004:** Calibration results for different data and registration methods.

Matching Algorithm	Name	Method I (°)	Method II (°)	Method III (°)	Average Error (°)
NDT	Rotation	4.16	3.29	3.06	3.50
	Park	11.91	9.64	12.77	11.44
	Campus	2.31	1.80	1.89	2.00
	Walk	4.97	4.19	5.65	4.94
	Cnu	16.13	3.00	9.60	9.58
OMP-NDT	Rotation	4.37	3.41	3.68	3.82
	Park	11.70	11.01	12.03	11.58
	Campus	1.49	1.36	0.84	1.23
	Walk	3.09	2.87	3.17	3.04
	Cnu	15.27	2.76	10.59	9.54
ICP	Rotation	4.98	3.55	3.40	3.98
	Park	17.24	14.23	14.29	15.25
	Campus	2.08	1.48	1.53	1.70
	Walk	12.27	8.77	6.08	9.04
	Cnu	17.79	6.87	4.17	9.61
GICP	Rotation	4.26	2.39	2.45	3.03
	Park	11.16	8.11	8.16	9.14
	Campus	1.91	1.31	1.35	1.52
	Walk	8.11	6.65	4.44	6.40
	Cnu	12.45	6.17	4.26	7.63

**Table 5 sensors-23-03119-t005:** Average angular distance of different data.

Name	Average Angle Change (°)	Average Calibration Error (°)
Park	2.23	11.85
Walk	2.30	5.86
Rotation	3.79	3.58
Campus	3.06	1.61
Cnu	0.99	9.09

## Data Availability

The data presented in this study are available in [15]. To facilitate the research community, the project is openly available at https://gitee.com/yinshilun/two-step-calib.

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
