# Peer review of "Uncontrolled Two-Step Iterative Calibration Algorithm for Lidar–IMU System"

_sensors, 2023, doi:10.3390/s23063119_

Round 1

Reviewer 1 Report

This study presents a new uncontrolled two-step iterative calibration algorithm that eliminates motion distortion and improves the accuracy of the lidar- imu system. The algorithm first corrects for rotational motion distortions by matching the original inter-frame point cloud.

Both in the abstract and in the experiments, the authors mention that the method in this study produces significantly better calibration results for vehicle-mounted, handheld, backpack, and other LiDAR acquisition methods compared to traditional methods. The algorithms have high accuracy, robustness and efficiency compared to existing algorithms. However, both the method and the experimental discussion seem to me to be based on improvements rather than completely new. And, is based on a comparison of algorithms that exist in existing databases. If it is only a test comparison, this is not wrong, but using this to suggest that this method is more applicable to multiple Lidar is not accurate and not strongly convincing.

I think it is interesting that "In our future work" is a real innovation that I would like to see. Only a multifactorial and multifaceted experimental analysis of complex systems can lead to a new SLAM algorithm that is different from the original database algorithm or other algorithms. Therefore I suggest to add to the original manuscript. Of course, this is not a requirement, but rather a reflection of where the real innovation of the manuscript lies.

Reviewer 2 Report

This paper presented a novel algorithm for sensor calibration. It is pretty important in the robotic area. Overall, the structure of this paper is well organized, and the presentation is relatively clear. The idea is interesting and has potential. However, there are still a few problems that need to be carefully addressed before a possible publication. More specifically,

Q-1 In the introduction, the importance of sensor calibration should be highlighted. For example, in the area of autonomous driving, it could improve the state estimation including position, velocity, and attitude. Thus, it would enhance the performance of vehicle safety control. Some related references to the drivetrain control applications should be discussed in the introduction: a hierarchical energy efficiency optimization control strategy for distributed drive electric vehicles; dynamic drifting control for general path tracking of autonomous vehicles; automated vehicle sideslip angle estimation considering signal measurement characteristic.

Q-2 For the calibration of the sensor, time synchronization and sampling frequency would also have an effect on the results. For example, we always assume that IMU has the ability of high frequency and no delay. On the contrary, GNSS, LiDAR, and camera would be the low frequency with signal delay. Some related work should be added in the introduction: visionaided intelligent vehicle sideslip angle estimation based on a dynamic model; IMU-based automated vehicle body sideslip angle and attitude estimation aided by GNSS using parallel adaptive Kalman filters.

Q-3: In Figure 7, please draw with different line types.

Q-4: Optimize Figure 2 to align with the page width.

Round 2

Reviewer 1 Report

The authors have read my review carefully and have made the appropriate changes in their response as well as in the revised draft. The revised work has met with my approval and I am therefore pleased to recommend this manuscript for acceptance in its current form.